# Development of an Image Analysis Pipeline to Estimate Sphagnum Colony Density in the Field

**DOI:** 10.3390/plants10050840

**Published:** 2021-04-22

**Authors:** Willem Q. M. van de Koot, Larissa J. J. van Vliet, Weilun Chen, John H. Doonan, Candida Nibau

**Affiliations:** 1National Plant Phenomics Centre, Institute of Biological, Environmental and Rural Sciences, Aberystwyth University, Aberystwyth SY23 3EB, UK; wqv@aber.ac.uk (W.Q.M.v.d.K.); lav4@aber.ac.uk (L.J.J.v.V.); jhd2@aber.ac.uk (J.H.D.); 2Faculty of Electrical Engineering, Mathematics and Computer Science, Technical University Delft, 2628 XE Delft, The Netherlands; w.chen-6@student.tudelft.nl

**Keywords:** *Sphagnum*, peatmoss, capitula density, image analysis, computer vision

## Abstract

*Sphagnum* peatmosses play an important part in water table management of many peatland ecosystems. Keeping the ecosystem saturated, they slow the breakdown of organic matter and release of greenhouse gases, facilitating peatland’s function as a carbon sink rather than a carbon source. Although peatland monitoring and restoration programs have increased recently, there are few tools to quantify traits that *Sphagnum* species display in their ecosystems. Colony density is often described as an important determinant in the establishment and performance in *Sphagnum* but detailed evidence for this is limited. In this study, we describe an image analysis pipeline that accurately annotates *Sphagnum* capitula and estimates plant density using open access computer vision packages. The pipeline was validated using images of different *Sphagnum* species growing in different habitats, taken on different days and with different smartphones. The developed pipeline achieves high accuracy scores, and we demonstrate its utility by estimating colony densities in the field and detecting intra and inter-specific colony densities and their relationship with habitat. This tool will enable ecologists and conservationists to rapidly acquire accurate estimates of *Sphagnum* density in the field without the need of specialised equipment.

## 1. Introduction

In recent years, there has been a renewed interest in the study and restoration of peatlands. Although peatlands only cover 2–3% of the earth’s surface [1] they store around a third of the world’s soil carbon. Overextraction of peat and government subsidised bog drainage for agricultural purposes have led to the degradation of many peatland ecosystems. In addition, changes in global climate patterns, especially precipitation, have accelerated peatland degradation in some regions [2,3]. Peatlands are a critical part of the global carbon cycle and their conservation and restoration is key if we are to meet global climate targets [4].

Key to conservation efforts is understanding the interactions between the plant communities in the peatland ecosystem and their responses to environmental changes [5,6,7,8]. A main component in many peatland ecosystems is the peatmoss, *Sphagnum*, which is often given the status of ‘ecosystem engineer’ [9,10,11,12]. Many of the species in this genus are bog specialists that waterlog and acidify the environment, excluding many potential competitors and creating optimal peat-forming conditions. *Sphagnum* mosses possess specialised morphological and physiological adaptations for water retention, which promote carbon sequestration through peat accumulation. The photosynthetically active tissues of *Sphagnum* mosses are located at or near the surface and each shoot of *Sphagnum* bears a dense array of growing branches at the top called the capitulum. In the capitulum region, branches elongate much more rapidly than shoot internodes, producing the characteristic compact hemispherical shape. Below the capitulum region, the branches tend to stop growing and internode elongation increases. Further down the stem, the lack of light causes the death of the lower branches and, characteristic to *Sphagnum* mosses, the rate of degradation of the dead material may be so low that it allows the formation of peat. The peat layer plays an important role in carbon sequestration, water storage, and prevention of surface desiccation. The acidic waterlogged conditions in the peat can slow or prevent the growth of most other plants and microorganisms [13]. Despite its importance in the maintenance of the peat ecosystem, we are only starting to understand how *Sphagnum* growth and development responds to environmental changes, specifically to drought.

The Sphagna are not desiccation tolerant (DT) and are generally restricted to wet and humid habitats. However, different niche preferences between the species have been linked to the micro-climate and, specifically, to the accessibility of ground water [11]. Some species are specialised, but others can grow across a range of habitats. The basis of these preferences and adaptations to different habitats remain poorly understood. Generally, it is accepted that *Sphagnum* avoids desiccation by a combination of having a large water storage capacity, the ability to retain water within organ spaces, and strong capillary forces that replenish the surface evaporated water from the peat layer below [14,15]. *Sphagnum* holds large amounts of water, perhaps more per unit of biomass than any other group of land plants. Some of the water is stored in specialised cells, but the majority is retained within extracellular spaces [14]. The ability to store water in extracellular spaces is due to morphological adaptations of the *Sphagnum* plant, including densely packed leaves and branches, and the production of long pendant branches that act as wicks [9,16]. In other bryophytes, colony density has been shown to have a significant impact on water retention by reducing the amount of free space between individual shoots and trapping moisture below the canopy [9,17,18]. Although this has also been suggested for *Sphagnum*, there is little experimental data. The density of the canopy can be defined as the number of capitula per unit area, but the arrangement of leaves and branches also varies and may contribute to water relations. Species-specific canopy traits could be plastic and change according to the habitat or alternatively, canopy traits could influence the habitats that a given species colonises [19,20]. 

Canopy density is not only important to determine drought tolerance, but is also a critical determinant in the successful establishment of peat plugs in bog restoration projects. Many of the natural peatlands in the UK have been subjected to extensive human intervention and are now considered to be degraded. Despite this, bogs are now recognised as important and diverse habitats and many restoration projects are under way across the UK. Although *Sphagnum* re-introduction has been mostly done using mixed plugs [21], optimal densities and combinations of species are not known. Estimation of *Sphagnum* mosses colony density in the field is a time-consuming process and as far as we know, easy-to-use protocols are not yet available.

Methods for estimating canopy density are well established for trees and other large plants, but few have been applied to bryophytes [22,23,24]. These methods exploit a range of imaging techniques, including aerial photography, satellites, and laser scanning. Data from these methods ranges from simple density measures to the physiological status of the plants using various vegetation indices. In recent years, there has been an upsurge in the development of imaging techniques being applied to plants. Increased image resolution and advances in image analysis have facilitated monitoring growth and development [25,26,27,28] and are easily applied to both small and large scale surveys. However, many imaging techniques require specialised equipment such as sophisticated imaging platforms that cannot easily be used in the field, or access to unmanned aerial systems, which often do not provide the necessary detail for many studies. High-resolution cameras have become ubiquitous on smartphones, and these have great potential for the collection of imaging data through citizen science. The development of appropriate analytical tools could increase the monitoring capacity of conservationists. In addition, such tools could enable scientists to rapidly acquire large amounts of data and help standardise data collection and analyses. 

Here, we describe the development of a simple, robust, and user-friendly image analysis pipeline to determine *Sphagnum* canopy density in the field using images acquired with a standard smartphone. We show the accuracy and applicability of the method for images acquired in different sites and for different *Sphagnum* species. Using the developed pipeline, we show that densities of several *Sphagnum* species are different in the field and quantify how densities differ within the same species growing in different sites.

## 2. Materials and Methods

### 2.1. Site Selction 

*Sphagnum* species from several different local ecosystems were selected to cover the four most abundant taxonomic sections within the genus, namely *Sphagnum*, *Acutifolia*, *Subsecunda,* and *Cuspidata* (Table 1) and to provide a reasonably representative mixture of the sizes and shapes that are typically found in the genus. *Sphagnum* colonies were imaged at three field sites, Coed y Darren (SN677835), Pen y Garn (SN791758) and Llyn Pendam (SN709838) (Figure 1). The selected sampling sites covered a range of *Sphagnum* niches, such as wet woodland, minerotrophic mire, and blanket bog. Within each site, several patches of each moss species were selected. In order to encompass as much morphological, physiological and imaging diversity as possible, images were acquired across these sites in different seasons and under different weather conditions.

### 2.2. Field Image Acquisition

Images were acquired with a simple, purpose-built imaging rig that is easy and inexpensive to build, portable and flexible enough to allow the use of different smartphones or cameras (Figure 2). The rig consisted of a rectangular imaging area (31 × 26 cm) delineated by a blue surrounding that held a colour card (greywhitebalancecolourcard.co.uk) and supported a 35 cm high platform on which the smartphone was placed (Figure 2A,B). The blue frame facilitates easy removal from the image, as blue is a distinct colour not usually found in nature, and ensures a standard field of view (FOV) for each acquisition. The presence of a commercial colour card was used as a landmark and to aid white balance correction. To ensure general utility, both Apple (iPhone 6 s and 7) and Samsung J3 smartphones were utilised. The rig was placed on a selected patch of *Sphagnum*, making sure that the lighting on the region of interest (ROI) was homogenous. The imaging sites were selected based on the presence of appropriate types of moss but also areas with low amounts of vascular plants, as they might obstruct the imaging. Special care was paid to making sure that the whole frame was seen in the picture, that the view of the colour card was not obstructed, and the image was not overexposed (Figure 2C). A black plastic screen was used to block direct sunlight and ensure homogeneous illumination. A total of 82 images were collected across the different sites and from different days, 30 from Pen y Garn, 29 from Llyn Pendam, and 23 from Coed y Darren (Table 1). All images used are available at https://doi.org/10.20391/9ba4df8f-2d28-4bce-b9b8-ec6368b636e0 (accessed on 21 April 2021).

### 2.3. Image Processing Pipeline

A computer vision and image processing pipeline was developed in Python 3 [29] (http://www.python.org) (accessed on 21 April 2021) and is available, together with instructions for use at http://www.github.com/NPPC-UK/Capitula-Counter (accessed on 21 April 2021) as a Python file. Image blurring, colour conversion, and cropping were carried using the OpenCV package [30] and exposure compensation was done using the PlantCV package [31]. The interactive GUI for thresholding in the HSV (Hue, Saturation, and Value) S colour channel (used for masking non-*Sphagnum* objects in the image) was developed using the PySimpleGUI package [32]. Capitula detection was performed in the YCrCb Cb colour channel, where Y is luma and Cb and Cr are Chroma blue and Chroma red respectively, using the Difference of Gaussian Blob detection function from the scikit-image package [33]. This function blurs an image multiple times with increasing standard deviations and stacks the resulting blurred images into a cube. In this cube, local maxima of pixel values are detected as concentrations of high values, which detects them as ‘blobs’ in the image. In our case, the contrast between the background material and the capitula enables the function to find the capitula as local maxima that are then annotated as blobs. The annotations were saved as .jpg image files and listed in a .csv file with their positional coordinates and dimensions. The number of annotations per image was written to a separate .csv file containing a list of all the images. All the image annotation raw data is available at https://doi.org/10.20391/9ba4df8f-2d28-4bce-b9b8-ec6368b636e0 (accessed on 21 April 2021).

### 2.4. Thresholding Channel Selection

To determine the optimal colour channels for thresholding and annotation, a subset of 19 images was thresholded and annotated in a variety of colour-space combinations. Images were manually thresholded in five different commonly used colour space channels, namely HSV H, HSV S, LAB (where L is lightness, A is the green/red chromatic axis and B the blue/yellow axis) A, YCrCb Y, YCrCb Cr, as well as three mathematically constructed image representations: Chromatic difference, HSV difference and Green difference (Appendix A). Chromatic difference has been previously described [26,28]. Two new colour spaces, HSV difference and Green difference were designed to increase the contrast between the capitula and the background, by amplifying the channels (HSV H and RGB Green, respectively). The equations used to calculate HSV difference and Green difference can be found in Appendix A. All the resulting 152 images (19 images thresholded in the eight different colour spaces) were manually checked for the quality of the masks. The images with poor segmentation, where the distinction between *Sphagnum* and non-*Sphagnum* material was not clear, or where too much *Sphagnum* was also masked, were discarded. Forty-eight images from the different channels (HSV H (*n* = 6), HSV S (*n* = 15), LAB A (*n* = 2), YCrCb Y (*n* = 1), YCrCb Cr (*n* = 3), Chromatic difference (*n* = 4), HSV difference (*n* = 13) and Green difference (*n* = 4)) were taken forward and annotated in three different colour channels: LAB B, YCrCb Cb inverted, and HSV S (Appendix A), resulting in 144 annotated images. Each image was divided into 16 grid cells of equal size. For each image, four randomly selected grid cells were scored for true positives (correctly identified capitula), false positives (incorrectly identified as capitula), and false negatives (capitula not identified by the annotation) using the FIJI software (https://fiji.sc/) (accessed on 21 April 2021). The obtained scores were used to calculate precision, recall, and F-measure [34] using the following formulas:$$\mathrm{Precision}=(\frac{\mathrm{Count}-\mathrm{False}\;\mathrm{positives}}{\mathrm{Count}})$$
$$\mathrm{Recall}=(\frac{\mathrm{Count}-\mathrm{False}\;\mathrm{positives}}{\left(\mathrm{Count}-\mathrm{False}\;\mathrm{positives}\right)+\mathrm{False}\;\mathrm{negatives}})$$
$$F-\mathrm{measure}=(2*\frac{\mathrm{Precision}*\mathrm{Recall}}{\mathrm{Precision}+\mathrm{Recall}})$$

The higher the F-measure, the more accurate the identification of the *Sphagnum* capitula is. Images that had an F-measure lower than 0.60 or higher than 0.85 were reviewed by a second counter to confirm the counts. 

### 2.5. Pipeline Validation 

To verify the accuracy of the automated counts, a subset of 24 images from the field was manually counted independently by four individuals using the cell counter function in Fiji [35] and F-measures were calculated as described above.

### 2.6. Statistical Analysis

All statistical analyses were carried out in RStudio version 4.0.3 [36] using the packages car [37] and ggpubr [38]. Data and file manipulation was done in Python 3 [29] using the glob [29], numpy [39], and pandas [40,41] packages in Python. Figures were generated using Matplotlib [42] and seaborn [43] in Python. Comparisons between colour channels, species and counts were done through multivariate type III ANOVAs with Wilcoxon rank sum tests as pairwise comparison, as the data was unbalanced. Site comparisons were tested using Tukey’s honest significant difference test. Information on the statistical tests used for each analysis is included in the figure legends.

## 3. Results

### 3.1. Development of an Image Processing and Analysis Pipeline

In order to accurately measure colony density from the images acquired in the field, we developed a capitula detection and counting pipeline using Python. This pipeline and instructions for use are freely available at the NPPC GitHub (http://www.github.com/NPPC-UK/Capitula-Counter) (accessed on 21 April 2021) as a Python file. A schematic representation of the pipeline used for image processing and analysis is depicted in Figure 3.

The pipeline consists of two sections: Pre-processing and Annotation. The main steps in the pre-processing section are ROI detection, rotation, perspective correction and size standardisation, white balance correction, and finally cropping to the centroid. The cropped centroid then enters the annotation section where the image is first transformed to the HSV S colour channel, and thresholded to generate a mask. The masked image is then transformed to the YCrCb Cb colour channel, in which it is then annotated. Each step of the pipeline is described in detail in the following sections.

#### 3.1.1. Pre-Processing Pipeline

The first step in the segmentation pipeline used the blue frame to detect and select the image area that it is to be analysed (ROI) (Figure 3). Initially, the image is blurred using the blur function from the OpenCV package and a 50 × 50 kernel. This step is necessary to reduce noise caused by grass located outside the blue box or covering the edges of the blue box. Subsequently, the original image is converted to the HSV-colour space using the colour conversion function from OpenCV, which is the function used for all colour conversions in this pipeline. The blur and the HSV-colour space facilitate thresholding of the blue frame to produce the image mask, which is then eroded to reduce noise caused by stray pixels. The mask is filtered, and the four corners of the square located. To increase the robustness of this method the position of the lower right corner was calculated using the other three corners and knowledge of the frame shape. This considers unexpected blue pixels outside the frame or inadvertent differences in the camera orientation. The position of all four corners of the frame was then fed into the ‘getPerspectiveTransform’ function from OpenCV that calculates the transformation matrix to correct the perspective. The ‘warpPerspective’ function was used in turn to crop the image, correct the perspective, and transform the dimensions to a standard size. The resulting output, where the inside edges of the frame define the portion of the image, was used for downstream analysis.

Although exposure correction was not essential for estimating canopy density, it might be useful if the images are to be subsequently used to measure other parameters, such as pixel colour or saturation. We should note that over-exposed images cannot be easily corrected due to information loss. A white balance correction step using the colour card and the PlantCV package [31] was included (Figure 3) and as the colour card is always in the same position in our images, the positional coordinates (in our case 390x, 250y) were used and fed into the white balance function of the PlantCV package. This significantly reduced computational time, was less error-prone and less noise sensitive. The Auto-Detect Colour Card function can be used to find the white colour chip in the card if coordinates are not available.

After white balance correction, the images were further cropped to remove the blue frame and the colour card (Figure 3). Since all the images had the same shape and size, the x and y-coordinates from which the image was cropped were always the same (121x, 351y), and produced a cropped centroid with a width and height of 800 pixels, which corresponded to a size of 347 cm^2^. These dimensions can be easily adjusted for different imaging area sizes.

#### 3.1.2. Thresholding Optimisation

The first step in the annotation pipeline is image thresholding, where the non-*Sphagnum* material is removed from the image. This is done by identifying the pixel value range containing the *Sphagnum* material and using the pixels outside this range to create a mask. The mask is then used to remove all the non-*Sphagnum* pixels from the image. As this step is critical for the success of the annotation, we developed an interactive GUI using the PySimpleGUI package [32]. The GUI allows intuitive manipulation of the desired value ranges. After the image is cropped, the user is prompted to adjust the threshold in the interactive GUI window. The upper and lower thresholds can be adjusted for each image in order to obtain an image with the highest possible contrast between the capitula and the other plant material. A threshold was considered good when it sufficiently removed large amounts of non-*Sphagnum* material, whilst only minimally affecting the centre of the *Sphagnum* capitula. However, for sets of images of similar lighting or if high throughput is required, the user has the option to select general thresholding values for all the images. For this, we created a pop-up window at the start of the pipeline, which asks whether the user wants the version with the GUI and manually threshold each image or enter specific thresholding values that will automatically be applied for all images. For the majority of our images, we found that the upper threshold was usually constant at 256 while the lower value varied with the colour of the material as indicated in Appendix A. To avoid manual thresholding, the values in this table can be entered at the start. In the case of sets of images with variable colour and exposure, a general lower threshold of 125 can also be used, as we show that it yielded similar results to manual thresholding (Appendix A).

#### 3.1.3. Thresholding and Annotation Channel Selection

The performance of the pipeline relies on two key factors. The first one is achieving a good contrast between the *Sphagnum* and background material, such as grasses, so a mask can be created, and the background material removed. This is done by using an appropriate thresholding colour channel to create a mask. The second is to achieve a good enough contrast between the individual capitula so they can be detected by the Blob DoG function as local maxima and annotated as *Sphagnum* capitula. This can be achieved by selecting an appropriate annotation colour channel. In order identify the more suitable thresholding and annotation channels, 19 images were thresholded in five different colour spaces (HSV H, HSV S, LAB A, YCrCb Y, YCrCb Cr, Chromatic difference, HSV difference and Green difference). The images with better thresholding were annotated in three colour spaces (LAB B, YCrCb Cb inverted and HSV S) (Appendix A), as detailed in the methods. F-measures, which reflect how accurate a classifier is, were used to assess the performance of the various channel combinations.

Our analysis of the performance of different combinations of thresholding and annotation channels found that the combination of thresholding channel HSV S and annotation channel YCrCb Cb resulted in a high mean F-measure (0.77) (Appendix A). Although the F-measure for thresholding channel HSV S was not significantly different from other thresholding channels within annotation channel YCrCb Cb, it had the largest sample size (*n* = 15) and was therefore considered the most reliable. The high performance of the YCrCb Cb as an annotation channel was maintained when the different species were considered separately (Appendix A). The full comparison of the different thresholding and annotation channels can be found in Appendix A.

#### 3.1.4. Blob Detection and Capitula Counting

The last step in the pipeline was identification of *Sphagnum* capitula. For this, images segmented in the HSV S channel and transformed into the YCrCb Cb channel were fed into the Blob DoG (Difference of Gaussian) detection of the scikit-image package, which is a popular, open access package under constant development. Blob DoG creates blurred copies of the original image and calculates the difference for each pixel between the blurred copies, allowing edge detection based on when the value of a pixel changes from positive to negative or from negative to positive. The identified blobs were then overlaid on the original cropped image as annotations (Figure 3). Finally, the annotations are saved as a list containing their coordinates and dimensions, with the number of rows equalling the total number of annotations in the image. The length of this list for each separate image was written to a separate spreadsheet as the number of plants counted by the pipeline. All these are provided as .csv files.

### 3.2. Estimation of Pipeline Accuracy

Having established that the HSV S and YCrCb Cb channels were optimal for thresholding and annotation, a new set of 34 previously unused images was run through the pipeline using these channels as the default. Together with the images that had been run through these channels in the pipeline development stage, the resulting in a total of 68 images (Appendix A), were scored for true positives, false positives, and false negatives to calculate the evaluative scores.

The pipeline achieves a mean F-measure of 0.77 which indicates that it correctly identifies the capitula correctly 77% of the time when all species are combined. When we looked at each species separately (Figure 4), the highest F-measures were achieved by images of *S. quinquefarium* (0.93), *S. fallax* (0.90), and *S. inundatum* (0.91) while *S. papillosum* had the lowest F-measure (0.50) suggesting that the method does not perform as well for this species (Figure 4). We should note that a low number of images were used for *S. inundatum* (*n* = 4) and the mixed colonies of *S. fallax* and *S. papillosum* (*n* = 4) (Appendix A), and this could affect their mean F-measures.

Automated and manual counts were compared to determine the accuracy of the software in the identification of *Sphagnum* capitula. For this, a subset of 24 images, consisting of five images of each species except for *S. inundatum*, of which only four images were available, were run through the software and also counted manually using Fiji by four counters.

As can been seen in Figure 5, there were significant differences between the number of capitula counted by each of the counters, underlining the difficulties in this process. All the manual counts were significantly lower than the automated counts (*p* < 0.0001 ****) (Figure 5) indicating that, although the pipeline was able to achieve high F-measures, it typically overestimates the number of plants in an image.

In order to determine if there was a correlation between the manual and automated counts, a model was created which contained the manual count, automated count, and species. For the model, we used the 64 images described above to calculate the F-measures. The manual count for these images was calculated by adding the true positives and the false negatives scored by the counters. Both linear and log models were fitted to the data, and these performed similarly based on the Akaike Index Criterion (AIC) and residual standard error (Figure 6, Appendix A). As the linear and log models were not significantly different form each other (*p* = 0.173), we decided to use the linear regression model as it was the least complex interpretation of the data. The linear model was highly significant (*p* < 0.001 ***, R^2^ = 0.63) and had a slope of 0.483 (Figure 6, Appendix A), indicating that there is a linear relationship between the automated counts and the manual counts. Furthermore, the model indicated that this linear relationship was maintained across species (Appendix A), and a Kruskal-Wallis test indicated that there was no significant difference between the manual counters (*p* > 0.32).

As there was a linear relationship between the manual and automated counts, the data obtained from the model was used to calculate a correction factor for the automated counts. This correction factor was calculated as the mean of the manual count divided by the mean automated count, resulting in a correction factor of 0.81. When this blanket correction factor was applied to the automated counts, they were no longer significantly different from the manual counts for *S. fallax*, *S. inundatum,* and *S. auriculatum* (Figure 7A). However, as *S. papillosum* and *S. quinquefarium*’s corrected counts were still significantly different from the manual counts, species-specific correction factors were also calculated (Appendix A). Using these species-specific correction factors, the adjusted automated counts were no longer significantly different from the manual counts for any of the species (Figure 7B).

Given these data, an extra step was added to the pipeline before the colony density estimation. This step included the blanket correction as default, but this can be changed to use the species-specific correction factor described here or independently calculated.

### 3.3. Estimation of Capitula Density in Field Images

In order to determine if there were density differences between species and if different habitats influence species density, we analysed images containing different *Sphagnum* species from different sites. The species-specific correction factor was used for each species. Indeed, we observed differences in plant density (number of plants per cm^2^) for the different species and for the different habitats (Figure 8).

*S. quinquefarium* had the highest mean density (1.34 plants/cm^2^), closely followed by *S. fallax* (1.67 plants/cm^2^) (Figure 8A). *S. inundatum* had the lowest mean density (0.74 plants/cm^2^). These densities are similar to the ones obtained by manual counts in the field (*S. fallax* (1.47 plants/cm^2^), *S. papillosum* (1.24 plants/cm^2^) and *S. inundatum* (0.65 plants/cm^2^)). There was no significant difference in density for *S. quinquefarium* with *S. fallax* and mixed stands of *S. fallax* and *S. papillosum.* Interestingly, mixed stands of *S. fallax* and *S. papillosum* seemed to have a density overlapping with both species separately. *S. inundatum* and *S. auriculatum,* two closely related species but imaged in two different habitats, were also significantly different.

We then looked at the effect of habitat on colony density for a given species. Both *S. fallax and S. papillosum* were found in Pen y Garn, a minerotrophic mire and Llyn Pendam, a lakeshore habitat. We observed significant differences in the density of *S. fallax* between the two sites, with higher density observed in the drier lakeshore habitat (Figure 8B). As for *S. papillosum* we observed a slight but not significant difference between both sites (*p* = 0.07) with the higher density observed at the Llyn Pendam as seen for *S. fallax* (Figure 8C).

Finally, to demonstrate the scalability of the method, a further 160 independent images of both *S. fallax* and *S. papillosum* were acquired from a previously unrecorded section of Pen y Garn at the far end of the bog. The images were acquired in one morning over a period of 4 h and were then fed to the pipeline. Even with manual thresholding for each image, the total analysis took less than 30 min, showing that the method has the capacity to acquire and analyse hundreds of images in a day. When default threshold values were used (125–256) instead of manually thresholding, the run-time of the 160 images through the pipeline on a regular consumer laptop was less than 5 min. The average densities observed for *S. fallax* (1.27 plants/cm^2^) and *S. papillosum* (0.90 plants/cm^2^) (Figure 9) were similar to those obtained previously for Pen y Garn (1.11 and 0.82 plants/cm^2^ for *S. fallax* and *S. papillosum* respectively) (Figure 8). These data show that our method can easily be scaled up and it is time- and cost-effective in relation to manual methods.

Taken together, our data show that the developed pipeline can be used to accurately assess *Sphagnum* colony density in the field and could detect significant species and habitat differences that may be relevant to ecological and conservation studies.

## 4. Discussion

Plant density varies between *Sphagnum* cushions and it is thought to affect evaporation rates [9,18,44]. However, tools for the quantification of density have not been developed and therefore this still requires either destructive sampling or visual counting, both of which are time consuming. The image analysis pipeline presented in this study provides a tool for *Sphagnum* researchers that can quickly, and with similar accuracy as a human, provide an estimate of plant density. The pipeline uses images captured in the field with minimal disturbance and allows for intra and inter-specific comparisons. Being image-based, the procedure could be scaled and easily adapted for citizen-science projects.

Our pipeline can analyse any reasonably exposed image acquired with a standard smartphone, although having a frame of defined size and colour and acquiring the image using lower exposure settings on the phone worked best. The frame design not only ensured that the phone was kept at a constant distance from the imaging area, but the presence of the blue frame facilitated positional information and the selection of the region of interest (ROI) for each image. The pipeline is also very user-friendly, requiring only minimal input in the form of file paths for source and destination directories, after which a pop-up window allows for the selection of the intuitive GUI window for thresholding, or the input of custom threshold values to run a uniform large image set.

Finally, the pipeline is robust. The images were collected periodically from marked patches across the 2020 season and so covered a range of drought states and colours that are typical for *Sphagnum* in the field. In addition, in some sites, the *Sphagnum* patches were variable in size or had a variable number of higher plants growing through them. Although one might expect that the presence excessive non-*Sphagnum* material might have been a problem, our results suggest that this did not have a significant influence on the final counts. During the precision analysis, we used a randomised grid selection for each image and while some grid cells had mostly *Sphagnum* material, others had large amounts of non-*Sphagnum* plants. Despite this, statistical analysis showed that there was a highly significant relationship between the number of annotations in the image and the number of annotations in the selected grid cells, indicating that these grid cells were representative for the entire image (data not shown). In both cases, the inclusion of an interactive easy to use manual thresholding step in the pipeline was a critical factor for performance.

Our analysis also showed that transforming images into HSV colour space for thresholding gave the best results, as the F-measures were consistently high for both annotation spaces LAB and YCrCb. The inclusion of a two-step colour transformation process in the annotation pipeline proved to be the most effective as the different properties of HSV and YCrCB made HSV more suitable for thresholding whilst YCrCb and LAB significantly outperformed HSV in annotation. Presumably, this is a result of HSV providing a better contrast between the *Sphagnum* and non-moss material, whilst YCrCb and LAB provide a better contrast between individual capitula. Similar to our results, HSV and YCrCb have been suggested to be among the best colour spaces for image classification in deep learning models [27,45], although the research into image processing methods such as these is still very limited for applied plant science.

A few recent studies have looked at *Sphagnum* density in the field and, using manual counts, found that increased shoot density positively affected photosynthetic capacity [20]. However, as our data shows, individual researchers can return significantly different counts of plants per area. Our image analysis pipeline eliminates this person-dependent variation and was able to detect distinct density differences between species, within species between habitats, and between species that are closely related (*S. inundatum* and *S. auriculatum*).

The highest densities were found for *S. fallax* and *S. quinquefarium*. In peatland restoration projects, species such as *S. fallax* are often favoured [46], usually unintentionally, due to their more competitive, pioneer nature compared to many other species. They are often attributed to a wider range of habitats and are more drought tolerant compared to typical bog species such as *S. papillosum* [5,11]. *S. quinquefarium* in this study came from the most ‘atypical’ *Sphagnum* habitat, as Coed y Darren is a humid woodland on a predominantly north-facing slope with no accessible water table. In this habitat, *S. quinquefarium* is entirely dependent on a combination of precipitation events and ambient humidity. In this context, it is plausible that a higher plant density is favoured for this species as a denser canopy could help retain water vapour on the forest floor.

The same consideration could be applied to *S. fallax*, although Pen y Garn and Llyn Pendam are more typical wet *Sphagnum* habitats. At these sites *S. fallax* grows in patches: in Pen y Garn, at the edges of hummocks of *S. papillosum* and at Llyn Pendam in a flush at the north side of the lake.

The low density found for *S. inundatum* at Pen y Garn may be related to its habitat. *S. inundatum* was found in pools which remain saturated throughout the year and as such, density would be irrelevant for humidity retention. Its closely related species, *S. auriculatum*, had a slightly higher density, despite being somewhat larger in size than *S. inundatum*. At Llyn Pendam, this species grows on the shore of the lake and is subjected to fluctuations in the water level, and even occasionally drying out in prolonged drought periods. Therefore, higher plant densities at intermittently dry sites such as Llyn Pendam, could be an advantage. However, *S. inundatum* was locally less common than the other species and we had the lowest number of images, so further sampling would be needed to support these findings.

The plant density of *S. papillosum* was surprisingly low, given that this species forms dense hummocks and mats. This could be explained by the difference in capitulum size between *S. papillosum* and the other species in this study. Typically, *S. papillosum* is fairly robust and their capitulum diameter can be much larger than that of the other species [14].

The density differences observed for the same species across two different sites could also be explained based on the water availability in each site. Higher capitulum densities were observed for *S. fallax* and *S. papillosum* at Llyn Pendam when compared to Pen y Garn. While Pen y Garn is a permanently wet blanket bog, with little variation in the water table, access to the water table at Llyn Pendam is more intermittent and partly dependent on runoff precipitation from the adjacent hills. Thus, higher plant densities could be favoured in the latter site. As these differences in density are possibly a result of the microclimatological variation between sites, this neglected variable may also be useful for peatland restoration projects. The density values for these species could help determine an optimal amount of plants to use in so-called ‘*Sphagnum* plugs’, which are planted on bare peat to restore the habitat [21].

We have shown that the *Sphagnum* annotation pipeline has the ability to capture density differences between species with a wide range of morphological variation and across habitats, making it a useful and easily appliable tool for field ecological studies. In addition, the magnitude and variation in the automated count is similar to those found manually by different people, indicating that this tool is equally reliable.

Besides its direct use in capturing, quantifying, and immortalising data on ecologically relevant sites, we envisage that the *Sphagnum* annotation pipeline will have other uses. As part of the counting process, our pipeline places annotations on the images that could be further utilised to rapidly delimit ROIs for co-registration with other modalities. For example, ROIs of capitula could be overlaid on images of the same subject site taken with multispectral, hyperspectral, or chlorophyll fluorescence imagers. The pipeline could be used to generating large, reliable datasets to train machine-learning algorithms to quantify biological and physiological traits in *Sphagnum* to set up a high throughput image monitoring framework.

## Figures and Tables

**Figure 1 plants-10-00840-f001:**
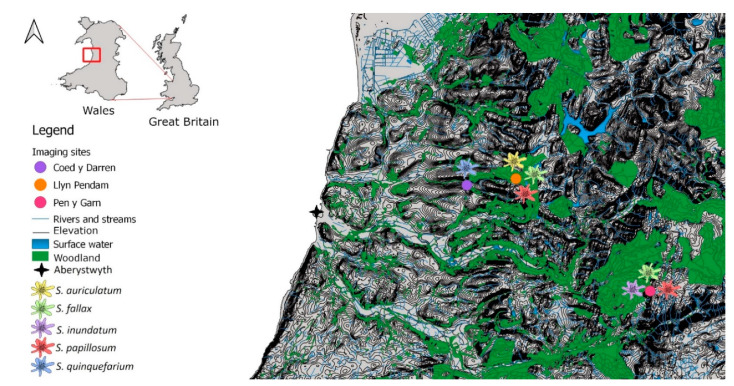
A map showing major surface water bodies, elevation lines, major woodlands, and the locations of the sampled sites around Aberystwyth, Wales and the species present at each site. Coed y Darren is a humid mixed woodland, with *S. quinquefarium* covering much of the forest floor. Llyn Pendam is connected to several other lakes by small streams, with mostly gravel shores with *S. auriculatum* growing on or near the shoreline, and *S. papillosum* growing near the in- and outflow streams. Pen y Garn is located in the Cambrian mountains with an ombrotrophic blanket bog covering its peak and ombrotrophic to minerotrophic mires at its base, with several pools that exhibit flow after heavy rain. Contains OS data © Crown copyright and database right 2020.

**Figure 2 plants-10-00840-f002:**
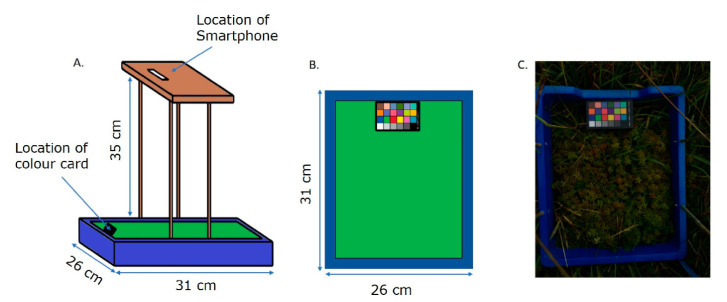
Schematic overview of the imaging rig. (**A**) Lateral view. (**B**) Top view. (**C**) Field image taken using the rig. The blue surround of the image facilitates removal in pre-processing, whilst the colour card provides an internal standard.

**Figure 3 plants-10-00840-f003:**
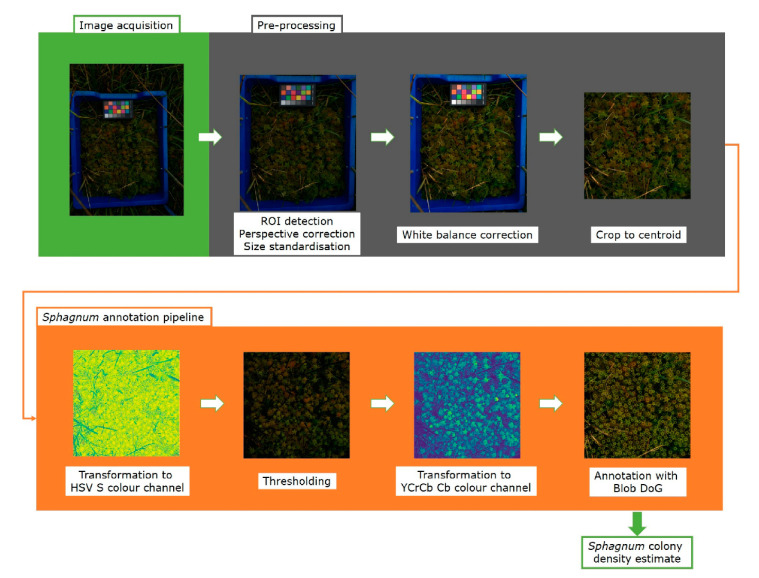
The capitulum counting image processing pipeline. After image acquisition, the images are pre-processed in Python using the OpenCV package, which involves detecting the blue frame, rotating it, and correcting the perspective. Then, using the PlantCV package, the colour card is used to correct the white balance of the image. The final step of the pre-processing crops the image to the 800 × 800 pixels centroid, removing the blue frame and colour card. After pre-processing, the image enters the annotation pipeline. First, the image is transformed to the HSV colour space channel S, in which it is subsequently thresholded to mask non-*Sphagnum* material. After thresholding, the image is converted to the YCrCb colour space channel Cb and the capitula are annotated using the Blob DoG function from the scikit-image package. From this, colony density is estimated and data is saved in a .csv file.

**Figure 4 plants-10-00840-f004:**
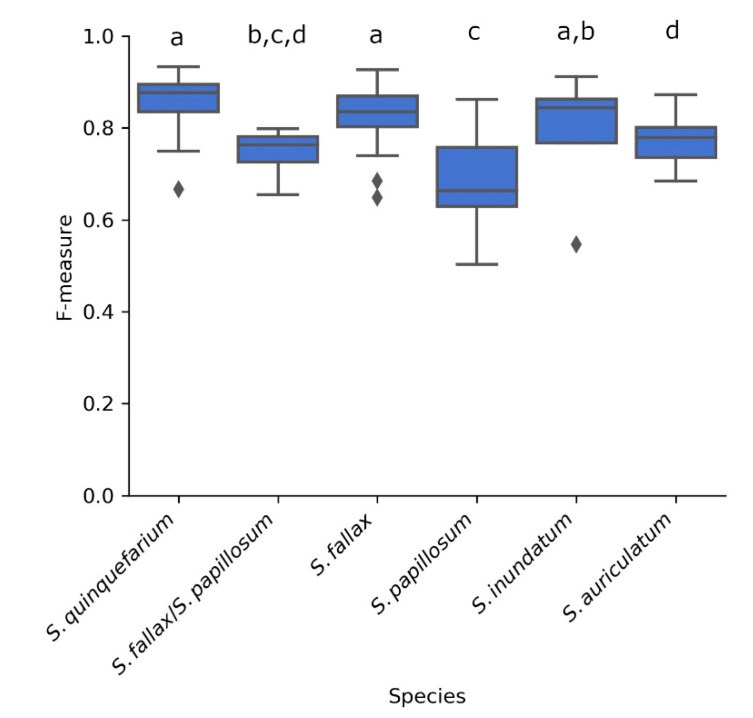
F-measures of the 68 images thresholded in HSV S and annotated in YCrCb Cb for each of the five species as well as mixed colonies of *S. fallax* and *S. papillosum* as indicated. Letters above the boxplots indicate significance groups for *p* < 0.01 as determined by Wilcoxon rank rum test.

**Figure 5 plants-10-00840-f005:**
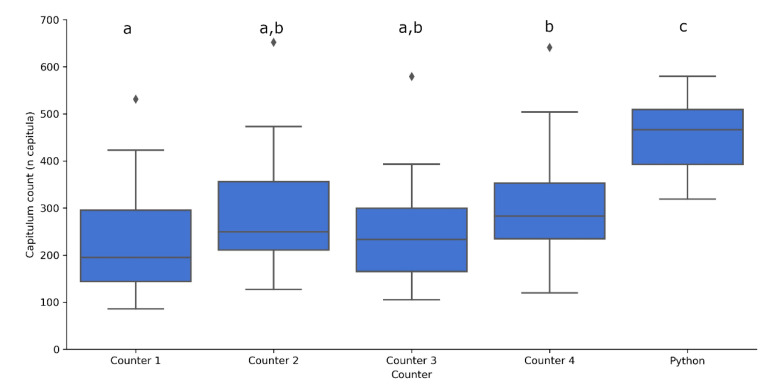
Number of capitula counted by the four human counters and the automated counting function in 24 images. The automated function was significantly different from the counters, and Counter 1 and Counter 4 differed significantly from each other. Letters above boxplots indicate significance groups for *p* < 0.05 between a and b and *p* < 0.0001 for c as determined by multivariate type III ANOVA with Wilcoxon rank sum tests.

**Figure 6 plants-10-00840-f006:**
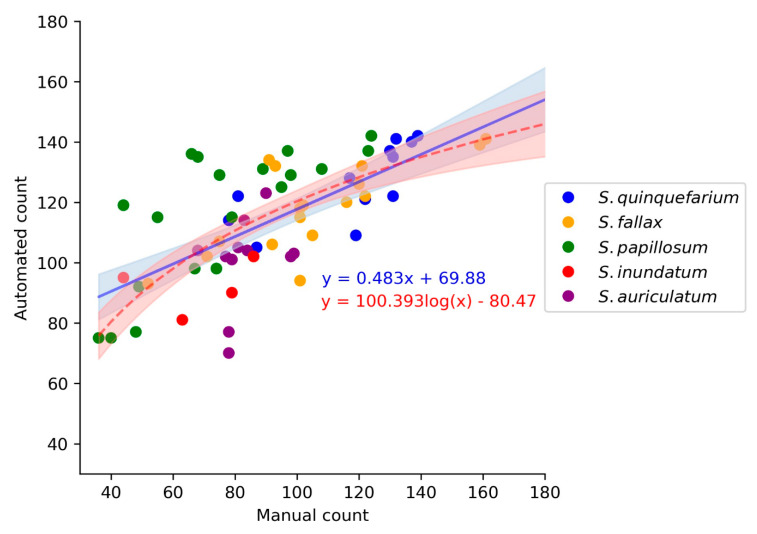
Comparison of the linear (blue line) and log (red dashed line) models for the manual count and automated count (*n* = 64). These models do not differ significantly (*t*-test, *p* = 0.173) and although the log model had a slightly lower AIC and residual standard error (Appendix A), we chose the linear model as it was the least complex way to explain the data.

**Figure 7 plants-10-00840-f007:**
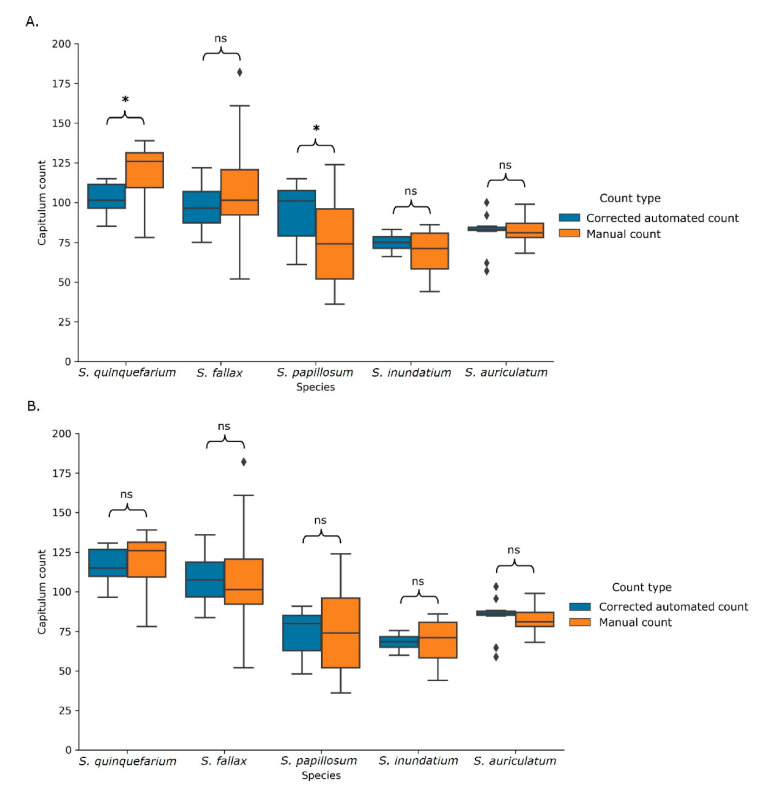
Blanket (**A**) and species-specific (**B**) corrected automated counts and the corresponding manual counts. Using the blanket correction (**A**), the corrected automated count still differed significantly for two of the five species. Using the species-specific correction (**B**) (Appendix A), there were no longer significant differences. ns indicates values not significantly different from each other and * indicates significance at *p* < 0.05 as calculated using a multivariate type III ANOVAs with the Wilcoxon rank sum test. Sample size per species can be found in Appendix A.

**Figure 8 plants-10-00840-f008:**
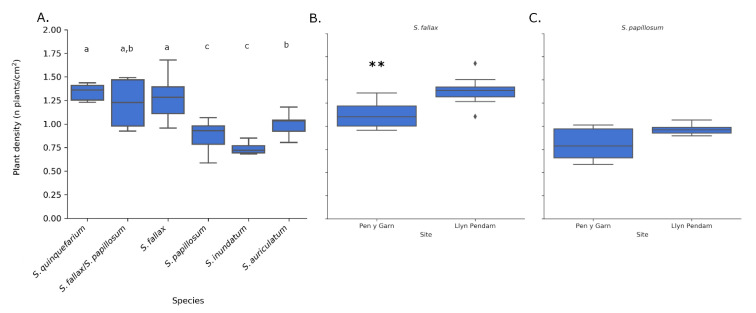
Plant density per cm^2^ for five species of *Sphagnum* (**A**) for all sample sites combined, using the corrected counts. There were significant differences (letters above boxplots indicate significance groups *p* < 0.05, Wilcoxon rank sum test) between the species and for *S. fallax* also between sample sites Pen y Garn and Llyn Pendam (*p* < 0.01 **, Tukey HSD test) (**B**). For *S. papillosum* the density was slightly higher at Llyn Pendam (**C**) but not significantly. Sample size per species and area can be found in Appendix A.

**Figure 9 plants-10-00840-f009:**
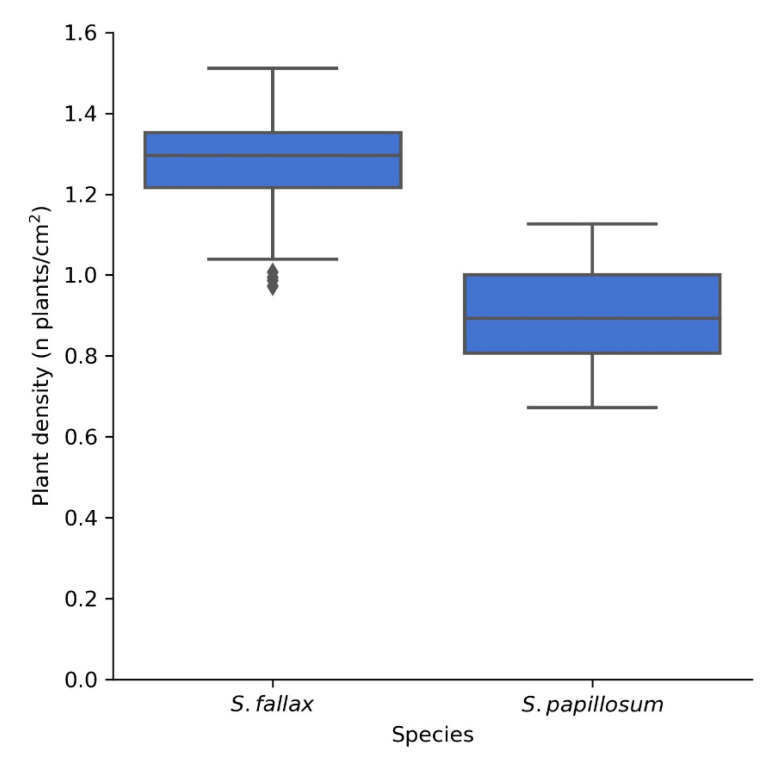
Plant density of *S. fallax* (*n* = 83) and *S. papillosum* (*n* = 77) from a previously unrecorded section of Pen y Garn. The images were taken over a period of 4 h and the analysis, when carried out manually, took less than 30 min. When fully automated, the analysis of these 160 images took less than 5 min, highlighting the potential for high throughput of this method.

**Table 1 plants-10-00840-t001:** Location, *Sphagnum* species, subgenus, and number of images of each species per location.

Location	Species	Subgenus	n. of Images
Coed y Darren	*S. quinquefarium* Warnstorf 1886	*Acutifolia*	23
Pen y Garn	*S. fallax* Klinggräff 1881	*Cuspidata*	9
	*S. inundatum* Russow 1894	*Subsecunda*	4
	*S. papillosum* Lindberg 1872	*Sphagnum*	12
	*S. fallax & S. papillosum* (mix)		5
Llyn Pendam	*S. auriculatum* Schimp.	*Subsecunda*	11
	*S. fallax*	*Cuspidata*	10
	*S. papillosum*	*Sphagnum*	8
Total			82

## Data Availability

All images used in this study are freely available at https://doi.org/10.20391/9ba4df8f-2d28-4bce-b9b8-ec6368b636e0 (accessed on 21 April 2021) and the software pipeline is freely available at the NPPC GitHub (http://www.github.com/NPPC-UK/Capitula-Counter/) (accessed on 21 April 2021).

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
