# Peer review of "Development of an Image Analysis Pipeline to Estimate Sphagnum Colony Density in the Field"

_plants, 2021, doi:10.3390/plants10050840_

Round 1

Reviewer 1 Report

This is a very interesting paper on a great use of widely available technology (smartphone cameras) to provide widely relevant ecological data (Peat cover). The paper will especially be of interest to anyone interested in peatlands, citizen science and applications of image analysis in ecology. The work is very original and is likely to be a strong contribution to the literature, especially in that it is a methods paper that proposes easy data collection and what looks to be a fairly smooth Python image analysis pipeline that could be used by many scientists (as well as teachers and citizen science proponents). I have noted below what I see as three significant issues that need to be addressed and then provide specific line by line comments below.

Major issues:

  1. An important consideration for a method like this is: How sensitive is this method to high cover and low cover (or many/few capitula)? I provided some specific suggestions in comments for Line 384 below, but there might be other equal or better approaches. Either way, if the purpose of the method is to automatically estimate a value (eg number of capitula) it is important to know whether the misclassification happens mostly on the low end, mostly on the high end or is randomly distributed across that span.

  1. The Methods (for thresholding and annotation) are generally confusing and hard to evaluate due to their opacity. Some simple improvements would include an overview paragraph (e.g. putting Fig 3 into words) at the beginning and also clear definitions of key terms that can mean different things in different contexts and lack clear definitions here (e.g. annotation, thresholding, grid cell). Given that this is a methods paper, the methods need to be very clear and they currently are not.

  1. The sample size seems very suspiciously low (<100). If the point of the method is automation and high-throughput, this tiny dataset feels unconvincing. It seems, based on the approach, that an observer ought to be able to go out and take several hundred photos in a day and then run them through the pipeline. Obviously there would be a constraint with the human counters in the validation step (for a subset of images perhaps), but a larger sample of images would be important if you were to convince readers that this could be scaled up to the extent that it is more time/cost effective than collecting data the old fashioned way. Given the amount of variation likely to be seen in nature, the small sample sizes here (especially when the number is partitioned across species and habitat types), are not as convincing as they could be with a more robust dataset.

Line 29: Subsidized, not subsided; purposes, not proposes

Line 31: “… have accelerated this process in some regions”- this needs a citation

Line 32: Grammatically incorrect sentence “Being a critical…”. There’s a subject/object issue involving the first clause of the sentence

Line 50: “ The peat layer is important…” grammatically incorrect sentence

Line 58: insert ‘species’…. “Some species…” to indicate that you’ve switched from niches to species in this sentence

Line 81: “country” instead of “county”?

Line 109: since these taxa are below the genus rank, check to see if they should be italicized

Line 131: perhaps “inexpensive” rather than “cheap”

Line 150 “total was” not “total were”

Line 218-  In this paragraph somewhere, you need to clearly state what the goal of thresholding is. You explain what you’re doing, but not why you’re doing it or what the actual or desired outcome of thresholding is in this context. My interpretation is that thresholding is the process of separating Sphagnum from non-Sphagnum pixels or parts of the image? I’m not confident and have trouble with the following paragraphs because it’s not clearly set up here.

Line 223- upper and lower thresholds must be adjusted manually? This impedes full automation

Line 238- in this paragraph- what is a “thresholding channel”? This is a critical point to understanding the methods and it’s not clearly explained

Line 252- More clarification needed. It is not clear what is going on at this stage. I assume that a reader familiar with image processing could follow this, but assuming the audience will be (like myself) biologists with limited image analysis expertise, more clarification will improve the chances of this method being widely used. I follow that 2/3 of the images did not pass the threshold, but there is no information about which images were selected (are certain color spaces more/less likely to pass threshold?). Also, it would be appropriate to define “segmentation” since this is a critical part of the visual thresholding. I assume it has a specific definition in this context and it would be useful to future users (and perhaps improvers) of the method to understand more clearly what is happening at this step.

Line 269- I did not follow the methods in this paragraph. It’s not clear what ‘grid cells’ are. Pixels? The scoring also lacks sufficient information. When the false positives etc were scored- it isn’t clear constitutes a true negative/positive or a false negative/positive and how that scoring was done. To be more specific, this implies that the investigators are comparing a modeled output against a visual interpretation, but it’s not clear (to me at least) what the modeled output is and also what exactly is being interpreted (what aspect of the “grid cell”- color value?).

Line 376- The wording here is confusing and could be clarified (The manual count…)

Line 379- I think it would be much more interesting to know whether the relationship holds up across different human observers than across Sphagnum spp

Line 384- This correcting approach needs further clarification, because the scalability of this method relies heavily on this correction. Is a simple linear correction best, for example, or should images with very many or very few captitula have different correction factors (i.e. is a simple linear regression really the best fit? The distribution of points in Fig 7 suggests that perhaps a curve ascending from the lower left and then flattening to the upper right might be a much better fit. This would provide a much different correction factor, and one that might be more broadly applicable to new images outside the training dataset

Line 408- grammatically incorrect sentence

Line 553 what kind of screen was used (size and material)?

Line 577- not clear what is meant by “… without compromising the capitual should be easier”.  Easier than what? Same in the following line- annotation should have fewer false positives than what?

Line 581- why were only 4 random cells per image used for the precision etc. Wouldn’t it be much better to use 1000 or more?

It’s not clear if this is being done by eye (by a person) or if these are being calculated. If it is arduous to do this, perhaps further clarification would be useful. As written, it’s not clear.

Author Response

We would like to thank the reviewer for finding our manuscript very interesting and their suggestions to improve it. We have addressed each of the comments below.

Major issues:

  1. An important consideration for a method like this is: How sensitive is this method to high cover and low cover (or many/few capitula)? I provided some specific suggestions in comments for Line 384 below, but there might be other equal or better approaches. Either way, if the purpose of the method is to automatically estimate a value (eg number of capitula) it is important to know whether the misclassification happens mostly on the low end, mostly on the high end or is randomly distributed across that span.

There is indeed a possibility that the method’s performance is less effective at low and high cover. However, in re-examination of our data, we found no evidence for this. As the reviewer suggested, we tested the log model and found that the linear model performs as well and now include this data in Figure 6 and the evaluative statistics as Table S4 and S5. Although we cannot rule out that there might be some issues when very low numbers of Sphagnum capitula are present on the image; in our lower density images the software was able to provide a good estimation of the actual density. At the upper end, the number of plants in an image is limited by spatial constraints in the patch of Sphagnum itself as there can only be so many plants per unit of area. In addition, the vast majority of images acquired in a typical Sphagnum habitat, will be in the linear part of the curve and thus within the high performance levels of the software.

  1. The Methods (for thresholding and annotation) are generally confusing and hard to evaluate due to their opacity. Some simple improvements would include an overview paragraph (e.g. putting Fig 3 into words) at the beginning and also clear definitions of key terms that can mean different things in different contexts and lack clear definitions here (e.g. annotation, thresholding, grid cell). Given that this is a methods paper, the methods need to be very clear and they currently are not.

As requested, we have simplified the methods description in section 2.2. We have moved some of the details to the methods section and simplified the descriptions. As suggested, we have also expanded on the initial paragraph of section 2.2 including a more complete description of Figure 3 and included explanations of the terminology used.

  1. The sample size seems very suspiciously low (<100). If the point of the method is automation and high-throughput, this tiny dataset feels unconvincing. It seems, based on the approach, that an observer ought to be able to go out and take several hundred photos in a day and then run them through the pipeline. Obviously there would be a constraint with the human counters in the validation step (for a subset of images perhaps), but a larger sample of images would be important if you were to convince readers that this could be scaled up to the extent that it is more time/cost effective than collecting data the old fashioned way. Given the amount of variation likely to be seen in nature, the small sample sizes here (especially when the number is partitioned across species and habitat types), are not as convincing as they could be with a more robust dataset.

We fully agree with the reviewer’s comments regarding the low number of images in the dataset. One of the reasons we did not include more images in the initial analysis is that during the time period that the work described in this manuscript was developed there were strict travel restrictions in place in Wales due to COVID-19. As such we did not have access to the different locations (all outside the local area) to acquire more images. These restrictions have now been relaxed so we were able to go to the Pen-y-Garn to collect more images. We have included an extra section and a new Figure 9 at the end of results where we describe the collection and analysis of these new images and use it to demonstrate the robustness and high-throughput of the method. We describe how is it possible to acquire and analyse 160 Sphagnum images in less than a day with accuracy comparable to what we had previously described.  This section demonstrates that the method can easily be scaled up and documents the time needed for the analysis.

Line 29: Subsidized, not subsided; purposes, not proposes

Corrected. 

Line 31: “… have accelerated this process in some regions”- this needs a citation

Relevant citation inserted. 

Line 32: Grammatically incorrect sentence “Being a critical…”. There’s a subject/object issue involving the first clause of the sentence

Corrected.

Line 50: “ The peat layer is important…” grammatically incorrect sentence

Corrected sentence now reads

‘The peat layer has important roles as a carbon sink, and as water storage, preventing the surface area from drying out.’

Line 58: insert ‘species’…. “Some species…” to indicate that you’ve switched from niches to species in this sentence

Corrected.

Line 81: “country” instead of “county”?

Corrected

Line 109: since these taxa are below the genus rank, check to see if they should be italicized

Corrected.

Line 131: perhaps “inexpensive” rather than “cheap”

Changed. (This section has been moved to Materials and methods as requested by reviewer 2)

Line 150 “total was” not “total were”

Changed. 

Line 218-  In this paragraph somewhere, you need to clearly state what the goal of thresholding is. You explain what you’re doing, but not why you’re doing it or what the actual or desired outcome of thresholding is in this context. My interpretation is that thresholding is the process of separating Sphagnum from non-Sphagnum pixels or parts of the image? I’m not confident and have trouble with the following paragraphs because it’s not clearly set up here.

Line 238- in this paragraph- what is a “thresholding channel”? This is a critical point to understanding the methods and it’s not clearly explained

Line 252- More clarification needed. It is not clear what is going on at this stage. I assume that a reader familiar with image processing could follow this, but assuming the audience will be (like myself) biologists with limited image analysis expertise, more clarification will improve the chances of this method being widely used. I follow that 2/3 of the images did not pass the threshold, but there is no information about which images were selected (are certain color spaces more/less likely to pass threshold?). Also, it would be appropriate to define “segmentation” since this is a critical part of the visual thresholding. I assume it has a specific definition in this context and it would be useful to future users (and perhaps improvers) of the method to understand more clearly what is happening at this step.

As part of the re-writing of section 2.2 we have clarified the goal of each step of the pipeline, including thresholding, thresholding channel choice and the criteria used to select the thresholded images. For each step we start by saying why we are doing it and then go on to explain how we did it. We explain why different colour channels were chosen and what makes a good colour channel for thresholding. We also describe in more detail the image selection procedure.

Thresholding is the act of limiting the total value range of an image to a new range within a threshold by changing the upper and lower boundaries of the value range. For example, an image usually has a range of 0-255. The distribution of values across this range can look like a normal distribution or other typical value distributions depending on the subject. Sphagnum, for example, on a smartphone acquired image, usually occupies a value range of between 125 and 255 in the colour channel HSV S. This means that all the values below 125 contain primarily non-Sphagnum information (grass, leaf litter, etc), and can be eliminated from further analysis by increasing the lower threshold to 125. As a result, all the pixels in the original value distribution that were below 125 are now ‘masked’, which essentially turns them into 0’s and removes them from the image. These thresholding processes are done in colour channels, which are different ways to represent the data in an image. The different colour spaces represent the pixel values in different ways and may accentuate different parts of the image better than the original RGB representation would. A good thresholded image would be one where the non-Sphagnum material is masked and the Sphagnum capitula left intact in the image. We now explain this in section 2.2.2 and 2.2.3.

Line 223- upper and lower thresholds must be adjusted manually? This impedes full automation

The upper and lower thresholds can be adjusted manually or set at a specific value (we provide values to use in different situations in Table S1) when high throughput is necessary. It is useful to maintain some flexibility and control within the pipeline to allow users to adjust it to their imaging conditions, but the software still performs well with the installed pre-sets. When the GUI first pops up, it provides the user with the option to use the manual thresholding sliders or to input a value range for when high throughput is required. When the latter is selected, the values put in by the user are used across all images and requires no further interaction from the user, meaning a large set of images can be run in minutes automatically. We explain this in lines 247-264. 

Line 269- I did not follow the methods in this paragraph. It’s not clear what ‘grid cells’ are. Pixels? The scoring also lacks sufficient information. When the false positives etc were scored- it isn’t clear constitutes a true negative/positive or a false negative/positive and how that scoring was done. To be more specific, this implies that the investigators are comparing a modeled output against a visual interpretation, but it’s not clear (to me at least) what the modeled output is and also what exactly is being interpreted (what aspect of the “grid cell”- color value?).

We have re-written this section and include clear details on why we are using grid cells and what the scoring process was. Lines 310-322 now read:

‘In order to determine, which channel performed best we manually scores the images for true positives (correctly identified capitula), false positives (incorrectly identified as capitula) and false negatives (capitula not identified by the annotation) using the FIJI software (https://fiji.sc/). The large number of images (114) and the need to use three different classifications, true positives, false positives and false negatives, made it impracticable to count all the capitula in each image. To circumvent this, and still obtain fully representative counts, each image was divided into 16 grid cells of equal size. For each image, 4 randomly selected grid cells were scored. The obtained scores were used to calculate precision, recall and F-measure…’

Line 376- The wording here is confusing and could be clarified (The manual count…)

We have re-written the sentence and now reads

‘All of the manual counts were significantly lower than the automated counts (p < 0.0001****) (Figure 5) indicating that, although the pipeline was able to achieve high F-measures, it typically overestimates the number of plants in an image.’

Line 379- I think it would be much more interesting to know whether the relationship holds up across different human observers than across Sphagnum spp

We checked if the manual counters differed from each other using a Kruskal-Wallis test, as the data was non-parametric and unbalanced. We found no significant difference between the counters (p>0.32).

Line 384- This correcting approach needs further clarification, because the scalability of this method relies heavily on this correction. Is a simple linear correction best, for example, or should images with very many or very few captitula have different correction factors (i.e. is a simple linear regression really the best fit? The distribution of points in Fig 7 suggests that perhaps a curve ascending from the lower left and then flattening to the upper right might be a much better fit. This would provide a much different correction factor, and one that might be more broadly applicable to new images outside the training dataset

We thank the reviewer for this comment, we tested the use of a log model instead of a linear one, and although the log model seemed to follow the trend of the data, slightly better as suggested by the reviewer, the model evaluative statistics indicated that this model did not perform significantly better than the simple linear model on our data (see Figure 6 and Table S4 and S5). As the linear model offers a simpler explanation of the data we decided to use it. We include all the information on both models in the text (lines 435-444) and in Figure 6 and Tables S4 and S5.

Line 408- grammatically incorrect sentence

We have re-written the sentence and now it reads.

‘In order to determine if there were density differences between species and if different habitats influence species density, we analysed images containing different Sphagnum species from different sites.’

Line 553 what kind of screen was used (size and material)?

Information now included.

Line 577- not clear what is meant by “… without compromising the capitual should be easier”.  Easier than what? Same in the following line- annotation should have fewer false positives than what?

We clarified what we mean in both cases. The two sentences now read:

‘The best contrasting channels allow masking of the majority of the non-moss material while keeping the Sphagnum capitula intact. This would improve capitula identification and decrease the number of false positives where non-moss material is incorrectly identified as a capitulum.’

Line 581- why were only 4 random cells per image used for the precision etc. Wouldn’t it be much better to use 1000 or more? It’s not clear if this is being done by eye (by a person) or if these are being calculated. If it is arduous to do this, perhaps further clarification would be useful. As written, it’s not clear.

We believe that there is a confusion of what the grid cells are. As requested above we have now clearly described what grid cells are and why they were used (lines 310-322). As the manual counting of true positives, false negatives and false positives is very laborious and slow and in order to increase the number and type of images counted, each image was divided in 16 regions of the same size using a grid. Each one of the resulting squares is considered a grid cell. On each image, 4 grid cells were chosen randomly and counts performed for those grid cells only.

Reviewer 2 Report

The manuscript entitled “Development of an image analysis pipeline to estimate Sphagnum colony density in the field” uses a novel and simple imaging system to estimate Sphagnum density at three sites in Wales. The paper presents the results of the flexible and easily implemented processing pipeline and compares the results to in-situ density measurements from the field. The results have implications for peat bog  management and restoration as well as citizen science initiatives. I find the method intriguing and the conclusions are overall valid but I have concerns about the analyses and presentation of results.

A main concern with the methods lies with the thresholding F-measures and ANOVA analysis. I do not fully understand what you are comparing or why with the type III ANOVA. The F-measure tells you the accuracy of the various thresholding/annotation techniques to correctly classify sphagnum and non-sphagnum pixels across the various species. You then put those F-measures along with species and the type of thresholding and annotation into an ANOVA to see how they compare (but you cannot do post-hoc tests due to the uneven data). I’m struggling to understanding the point of the ANOVA (other than to obtain p-values), particularly given the variable sample sizes and the fact that the F-measure is already a measure of performance. I am really not sure this is statistically valid or necessary. As I understand it you chose the HSV S channel based on the F-measure anyways, or did I misunderstand this? I think Section 2.2.3 could be restructured and simplified. Maybe not all results need to be presented here to improve clarity. I really think with a simplification of the thresholding results and substantial editing of manuscript figures, the paper will be greatly improved.

Below are detailed and general comments on how to improve the clarity and readability of the paper.

Abstract

Line 19: taken on different days

Introduction

Line 35: change to: ...efforts is understanding the interactions between …

Line 38: change attributed to appointed or given

Line 60: change to: …and adaptations to different habitats…

Line 60-61: add comma: generally, it is …

Line 62: add comma: … organ spaces, ….

Line 64-66: split this into two sentences. The current structure is confusing.

Line 75: change to: could be plastic

Line 81: remove: by

Line 94: remove: not always readily accessible – it’s not clear what this is referring to so I would suggest deleting it

Line 97-100: restructure and split into two sentences. Start the sentence with, “The development of appropriate analytical tools could increase monitoring capacity…”

Line 102-103: change to: acquired with a standard smartphone

Materials and Methods/Results….(?)

I am confused as to why the materials and methods comes after the discussion and how it is different than what is presented in what is called the results section. Please clarify this.

Line 108: if you want to separate Study Site into its own section you can do so

Line 107: This should be Materials and Methods, not Results….

Line 108-112: Split this into two sentences, too long.

Figure 1. It would be nice to also see a vegetation or land cover classification to the inset map if available to showcase the different niches. I would also suggest adding a label to the map of Wales and even including it within a UK map. From just looking at the figure as is, it is not clear where this is.

Line 132: change to: rig design

Line 134: change to: facilitates easy removal… and ensures a standard field of view (FOV) of each acquisition. – please correct me if I am wrong but that is the main point of the blue frame, right? Creates a standard FOV and is easily removed for image processing?

Line 140: change to: vascular plants

Line 140-142: This sentence is awkward with “making sure” 2 times. Please rewrite.

Line 152: change to: are available here:

Line 190-193: Change to: To increase the robustness of this method the position of the lower right corner was calculated using the other three corners and knowledge of the frame shape. This takes into account unexpected blue pixels outside the frame or inadvertent differences in the camera orientation.

- Try to improve the flow of your sentences by getting straight to the point and keeping them short. When you include too many ideas in one sentence separated by confusing commas (as was the case in the original construction) it decreases the readability and clarity of your writing.

Line 195 -198: Split this into two sentences: … to a standard size. The resulting output uses the inside edges…

Line 207: remove: of the PlantCV package

Line 227: change to: images with similar lighting or if high… - you don’t need to add the universal thresholding after the light conditions as this is made clear at the end of the sentence that this requires a general or universal threshold.

Line 238: remove: to use

Line 245: change to areas of interest – the plural comes only on the area, not interest

Line 247,253: When referring to equations, the plural of formula is formulas, not formulae

Line 249-251: Why do you provide the equations for these three and not the other 5 colour channels? They should at least be defined.

Line 255: change to: ROIs

Line 262: change to: were discarded

Line 263: change to: were used. Remove: It is worthwhile to note that

Line 265: I think you mean resulting sample size? Why are they so different and uneven? Is this because some of the images were overexposed?

Line 269: Don’t start a section with the word These. A total of 48 thresholded images. Reintroduce what you are talking about. It is also important to reiterate the goal of this section. To classify Sphagnum pixels. This is not written anywhere in this section. Make it easy for the reader to figure out what they are reading.

Figure 4: add a legend, I don’t know what the different colours mean. The x axis labels should include the entire threshold channel name. I am trying to use this figure to see where the HSV S is because this is what you end up choosing but it is really not clear to me. Also, this may just be my ignorance to the annotation and threshold channels, but can you please match the labels you use in the text to the labels in the figure. It makes interpretation very difficult for someone who is not super familiar with this nomenclature.

Line 280-281: for example, the F-measure was used to assess (not access) the performance of what?? For what purpose?

Line 308: what variables?

Line 323: where does the n=15 come from? This does not match any other n presented above.

Table 2. define SS in caption and Df.

What does this table tell us? We know

Line 351: change to: the method does not

Figure 5. what statistical test was used to get the p-values? If the uneven sample sizes influence F-measures, then a statistical comparison of the F-measure between species would seem to me to be invalid. If I am incorrect here please provide a better justification in the text (Line 352-353).

Figure 6. what statistical test was used?

Figure 7. Make the axes the same range 40-180 with a 1:1 line plus your trend line to better showcase the relationship and overestimation. Can you mark the species here so we can see that there is indeed no difference between species in the relationship?

Figure 8. I would try to combine the information in Table 3 and Figure 8. I think you could easily add a column with corrected count and note the significance there rather than have an additional figure. Also tell us what statistical test you used.

Line 408: what do you mean by we next applied to developed software? This is confusing, please clarify.

Figure 9. I am not sure I would call this adjusted density. This is the corrected estimated density from your pipeline right? Also in the caption, you are presenting density, not differences in density, remove: Differences in. You show these two species specifically because they were the only ones to show differences between sites? I would remake this figure to show all species in all sites (grouped box plot) and then note significant differences.

Line 420-421: this is a confusing construction. I would change it to the highest overall density.

Line 421-422: how can two species have the lowest mean density? I find this confusing as with the previous sentence.

Line 423: why isn’t it shown? Can you make it available in the Appendix at least?

Line 429: this sentence belongs in the methods, not results. I would remove it and get straight to presenting the results.

Line 436-438: this paragraph does not belong in the results. Please remove it or move it to the discussion where you can discuss it in detail.

Discussion

Line 440: It is best not to start sentences with an explanation, and then introduce what you are explaining after (especially in scientific writing). Start with what you are talking about in the sentence, namely plant density. Don’t make the reader wait until the end or middle of the sentence to figure out what you are talking about. The sentence reads much better like this: Plant density varies between Sphagnum cushions and is thought to affect evaporation rates. This sentence structure is common throughout and really reduces the readability and clarity of the paper. Please try to address this throughout the manuscript.

Line 442: destructive sampling

Line 443: what does this refer to? I would replace This with The image analysis pipeline presented in this study

Line 444-5: an estimate of

Line 443-446: split this sentence into two sentences. Too long as is.

Line 449-450: remove: , as the one we described here,  

Line 450-1: change to: worked best

Line 492: remove: as

Line 493: remove: as

Author Response

We thank the reviewer for considering our work to be useful for management and restoration processes. We have carefully considered each of the reviewer’s comments and provide detailed responses below.

A main concern with the methods lies with the thresholding F-measures and ANOVA analysis. I do not fully understand what you are comparing or why with the type III ANOVA. The F-measure tells you the accuracy of the various thresholding/annotation techniques to correctly classify sphagnum and non-sphagnum pixels across the various species. You then put those F-measures along with species and the type of thresholding and annotation into an ANOVA to see how they compare (but you cannot do post-hoc tests due to the uneven data). I’m struggling to understanding the point of the ANOVA (other than to obtain p-values), particularly given the variable sample sizes and the fact that the F-measure is already a measure of performance. I am really not sure this is statistically valid or necessary. As I understand it you chose the HSV S channel based on the F-measure anyways, or did I misunderstand this? I think Section 2.2.3 could be restructured and simplified. Maybe not all results need to be presented here to improve clarity. I really think with a simplification of the thresholding results and substantial editing of manuscript figures, the paper will be greatly improved.

We have used ANOVA to identify relationships between the different variables that might affect F-measure. We do agree with the reviewer that this may not be necessary and might make it more complicated to follow and understand the results. As such, we have removed the AVOVA analysis and description form the text and use it only in relation to the calculation of the significance values for the graphs. We have also re-written section 2.2 (see also comments from reviewer 1) and we believe that it is now clearer and easier to follow.

Below are detailed and general comments on how to improve the clarity and readability of the paper.

Abstract

Line 19: taken on different days

Corrected.

Introduction

Line 35: change to: ...efforts is understanding the interactions between …

Changed.

Line 38: change attributed to appointed or given

Changed.

Line 60: change to: …and adaptations to different habitats…

Changed.

Line 60-61: add comma: generally, it is …

Coma added.

Line 62: add comma: … organ spaces, ….

Coma added.

Line 64-66: split this into two sentences. The current structure is confusing.

Changed. Sentences now read:

Sphagnum holds large amounts of water, perhaps more per unit of biomass than any other group of land plants. Some of the water is stored in specialised cells, but the majority is retained within extracellular spaces [12].

Line 75: change to: could be plastic

Changed.

Line 81: remove: by

Removed.

Line 94: remove: not always readily accessible – it’s not clear what this is referring to so I would suggest deleting it

Removed.

Line 97-100: restructure and split into two sentences. Start the sentence with, “The development of appropriate analytical tools could increase monitoring capacity…”

We have restructured the sentences. They now read:

The development of appropriate analytical tools could increase the monitoring capacity of conservationists. In addition, it could enable scientists to rapidly acquire large amounts of data, and help to standardise data collection and analyses.

Line 102-103: change to: acquired with a standard smartphone

Changed.

Materials and Methods/Results….(?)

I am confused as to why the materials and methods comes after the discussion and how it is different than what is presented in what is called the results section. Please clarify this.

We have followed the journal’s instructions as to the structure and position the different sections. If the editors agree, we are happy to move the methods to after the introduction. The reasoning behind our structuring was to keep simple methodologies in the material and methods but if the methodology was specifically developed for this article, we expand on it in the results section. We believe that site selection, species choice, colour channel selection and thresholding methods were all considerations during method development and as such, we included them in the results section. We agree that the rig description and the considerations about the colour channels are better placed in the methods and have done so.

 Line 108: if you want to separate Study Site into its own section you can do so

Line 107: This should be Materials and Methods, not Results….

The description of the chosen study sites and the Sphagnum species used in this study are important to underscore the applicability of the method and we believe that placing it at the beginning of the results section makes this point clearer.

Line 108-112: Split this into two sentences, too long.

We have split this sentence in two as requested.

Figure 1. It would be nice to also see a vegetation or land cover classification to the inset map if available to showcase the different niches. I would also suggest adding a label to the map of Wales and even including it within a UK map. From just looking at the figure as is, it is not clear where this is.

We have re-done Figure 1. We now include a full Great Britain map and Wales is indicated in the map. We have also included vegetation and elevation in the map.

Line 132: change to: rig design

Changed.

Line 134: change to: facilitates easy removal… and ensures a standard field of view (FOV) of each acquisition. – please correct me if I am wrong but that is the main point of the blue frame, right? Creates a standard FOV and is easily removed for image processing?

Sentence changed and moved to Methods. It now reads

The blue frame facilitates easy removal from the image as blue is a distinct colour, not abundant in nature, and ensures a standard field of view (FOV) for each acquisition.

Line 140: change to: vascular plants

Changed.

Line 140-142: This sentence is awkward with “making sure” 2 times. Please rewrite.

Sentence now reads

Special care was paid to making sure that the whole frame was seen in the picture and that the view of the colour card was not obstructed.

Line 152: change to: are available here:

Changed.

Line 190-193: Change to: To increase the robustness of this method the position of the lower right corner was calculated using the other three corners and knowledge of the frame shape. This takes into account unexpected blue pixels outside the frame or inadvertent differences in the camera orientation.

Changed.

- Try to improve the flow of your sentences by getting straight to the point and keeping them short. When you include too many ideas in one sentence separated by confusing commas (as was the case in the original construction) it decreases the readability and clarity of your writing.

We have carefully gone through the manuscript and simplified sentences as requested to improve readability.

Line 195 -198: Split this into two sentences: … to a standard size. The resulting output uses the inside edges…

Changed.

Line 207: remove: of the PlantCV package

This section has been re-written.

Line 227: change to: images with similar lighting or if high… - you don’t need to add the universal thresholding after the light conditions as this is made clear at the end of the sentence that this requires a general or universal threshold.

Changed.

Line 238: remove: to use

Removed.

Line 245: change to areas of interest – the plural comes only on the area, not interest

Changed.

Line 247,253: When referring to equations, the plural of formula is formulas, not formulaeLine 249-251: Why do you provide the equations for these three and not the other 5 colour channels? They should at least be defined.

The sentence has been re-written and the equations moved to Table S2.

Line 255: change to: ROIs

Changed.

Line 262: change to: were discarded

Changed.

Line 263: change to: were used. Remove: It is worthwhile to note that

Changed and removed.

Line 265: I think you mean resulting sample size? Why are they so different and uneven? Is this because some of the images were overexposed?

The low number of images taken forward for some of the channels has to do with the contrast generated between capitula and the rest of the material being much lower in these colour spaces or the Sphagnum material being also masked. We now explain this process clearly in lines 293-297.

We have re-written the sentence

‘All the resulting 152 images (19 images thresholded in the 8 different colour spaces) were manually checked for the quality of the masks.  The ones with poor segmenta-tion, where the distinction between Sphagnum and non-Sphagnum material was not clear, or where too much Sphagnum was also masked, were discarded so that only 48 images from the different channels (HSV H (n=6 ), HSV S (n=15), LAB A (n=2 ), YCrCb Y (n=1 ), YCrCb Cr (n=3 ), Chromatic difference (n=4 ), HSV difference (n=13 ) and Green difference (n=4 )) were taken forward.’

Line 269: Don’t start a section with the word These. A total of 48 thresholded images. Reintroduce what you are talking about. It is also important to reiterate the goal of this section. To classify Sphagnum pixels. This is not written anywhere in this section. Make it easy for the reader to figure out what they are reading. Line 280-281: for example, the F-measure was used to assess (not access) the performance of what?? For what purpose?

Line 308: what variables?

As requested above, this section has been re-structured and simplified and these comments addressed.

Figure 4: add a legend, I don’t know what the different colours mean. The x axis labels should include the entire threshold channel name. I am trying to use this figure to see where the HSV S is because this is what you end up choosing but it is really not clear to me. Also, this may just be my ignorance to the annotation and threshold channels, but can you please match the labels you use in the text to the labels in the figure. It makes interpretation very difficult for someone who is not super familiar with this nomenclature.

The different colours are just arbitrary to refer to the different channels as indicated. We agree that they do not add any new information and have removed the colours from these and the other boxplots where colour is not informative. We have also revised the text and figures and for clarity use the full name for the thresholding and annotation channels.

Line 323: where does the n=15 come from? This does not match any other n presented above.

It is the number of images thresholded in the HSV S channel that were used for the annotation pipeline. We have clarified this in the text.

Table 2. define SS in caption and Df.

What does this table tell us? We know

As requested above we have removed the ANOVA analysis to the test and only retain the pvalues generated to discuss significance of the differences in the F-measure.

Line 351: change to: the method does not

Changed.

Figure 5. what statistical test was used to get the p-values? If the uneven sample sizes influence F-measures, then a statistical comparison of the F-measure between species would seem to me to be invalid. If I am incorrect here please provide a better justification in the text (Line 352-353).

Statistical test was multivariate type III ANOVAs with Wilcoxon rank sum test. Although for S. inundatum and the mixed fallax/papillosum we had low sample sizes, for the other species, although not all equal in number, there are enough representatives to draw conclusions on the general performance of the Blob DoG function on these species. We clarify this in the text lines 405-407.

Figure 6. what statistical test was used?

The information on what statistical test was used here and elsewhere was added to all the figure legends.

Figure 7. Make the axes the same range 40-180 with a 1:1 line plus your trend line to better showcase the relationship and overestimation. Can you mark the species here so we can see that there is indeed no difference between species in the relationship?

We have changed the axes to have the same ranges and coloured the data points for the species. This change does contribute to an easier interpretation of the data and we thank the reviewer for the suggestion. As for the extra 1:1 line we believe that its inclusion will make the figure interpretation more difficult especially now that, as requested by reviewer 1, we also included a log curve.  

Figure 8. I would try to combine the information in Table 3 and Figure 8. I think you could easily add a column with corrected count and note the significance there rather than have an additional figure. Also tell us what statistical test you used.

We have included information on the statistical test in the figure legend. We believe that combining the figure and the table makes it more complicated to interpret. We do agree that the information present in Table 3 is not critical to the understanding of Figure 8 so we have moved it to the supplemental information (Table S6).

Line 408: what do you mean by we next applied to developed software? This is confusing, please clarify.

This sentence was re-written it now reads

In order to determine if there were density differences between species and if different habitats influence species density, we analysed images containing different Sphagnum species from different sites.

Figure 9. I am not sure I would call this adjusted density. This is the corrected estimated density from your pipeline right? Also in the caption, you are presenting density, not differences in density, remove: Differences in. You show these two species specifically because they were the only ones to show differences between sites? I would remake this figure to show all species in all sites (grouped box plot) and then note significant differences.

We have changed the axis label and legends to plant density and removed differences in. The reason that we only show site differences for S. fallax and S. papillosum is because they were the only two species common to the three sites. The other three species (S. quinquefarium, S. inundatum and S. auriculatum) are only found at Pen y Garn.

Line 420-421: this is a confusing construction. I would change it to the highest overall density.

Line 421-422: how can two species have the lowest mean density? I find this confusing as with the previous sentence.

We have re-written these sentences to include only mean density values for the species, to avoid any confusion.

Line 423: why isn’t it shown? Can you make it available in the Appendix at least?

We now include this information in the text.

Line 429: this sentence belongs in the methods, not results. I would remove it and get straight to presenting the results.

We believe that for the interpretation of Figure 8B and C it is important to stress the differences in the habitats these plants come from to save the reader from having to go back to the beginning of the section and check.

Line 436-438: this paragraph does not belong in the results. Please remove it or move it to the discussion where you can discuss it in detail.

Although we agree that this paragraph is not strictly results we do believe that it helps to bring the results to a conclusion by summarizing the main findings so far and provides a link to the next section.

Discussion

Line 440: It is best not to start sentences with an explanation, and then introduce what you are explaining after (especially in scientific writing). Start with what you are talking about in the sentence, namely plant density. Don’t make the reader wait until the end or middle of the sentence to figure out what you are talking about. The sentence reads much better like this: Plant density varies between Sphagnum cushions and is thought to affect evaporation rates. This sentence structure is common throughout and really reduces the readability and clarity of the paper. Please try to address this throughout the manuscript.

We have re-written this sentence and edited the text in other places to increase readability and clarity as requested.

Line 442: destructive sampling

Corrected.

Line 443: what does this refer to? I would replace This with The image analysis pipeline presented in this study

Replaced

Line 444-5: an estimate of

Corrected

Line 443-446: split this sentence into two sentences. Too long as is.

Done.

Line 449-450: remove: , as the one we described here,  

Done.

Line 450-1: change to: worked best

Changed.

Line 492: remove: as

Removed.

Line 493: remove: as

Removed.

Round 2

Reviewer 1 Report

I feel that the authors have adequately addressed all of the concerns I had with the paper and they have done a very nice job of clarifying sections that were confusing in the first draft.

I am confident that this paper will be well received and congratulate the authors on a very interesting contribution.

Author Response

We would like to thank the reviewer for their supportive comments on our manuscript.

Reviewer 2 Report

I thank the authors for their initial review. Unfortunately, I still have some concerns about the organization of the manuscript and presentation of the results/methods that must be addressed. Additionally, the English still needs polishing throughout the manuscript (missing commas, grammatical errors, etc.). I provided extensive English editing in the first round some of which have not ended up in V2 of the manuscript (but are in the response document e.g. V2: Line 224, V1: line 227). I ask that the authors please carefully address this themselves in the second round.

Currently, the information about the development and validation of pipeline is split up between the results and methods resulting in a disjointed, resulting in an unclear presentation of the method. As this is a methods paper, this needs to be really clear to the reader (most of who will probably not be image processing experts).

As I said in my previous review, study site and image acquisition are never included in the results section. I understand that study site and image acquisition are relevant to the pipeline development, but I do not agree that they belong in the results section in this manuscript. The provided justification for me is not convincing and given there is an image acquisition section in the methods, I don’t really understand why it is also presented in the results. I ask that the authors clarify the organization of the sections among themselves and make a clear differentiation of what should be included in each section.

Additionally, the results section is confusing with repeated information (i.e., 2.2 and 2.2.1). I understand you are trying to provide a short summary in 2.2 before going into detail but then there are definitions and explanation that are introduced after acronyms are already used (i.e., HSV S, ROI) making the flow of this section difficult to follow. Skip the summary (2.2) or make it one or two general sentences and get straight into the details.

I have spent many hours reading the manuscript in detail and I want the method to come across as clearly as possible because it is valuable. I highly suggest the authors reorganize the results and methods as follows:

Results: Image processing pipeline and pipeline validation/accuracy

Methods: Study site, image acquisition, physical configuration/construction of the imaging rig, data analysis (validation method and statistical analyses)

Abstract

Line 10: remove: the

Line 11-12: and release of GHGs

Introduction

Line 29: Overextraction OR overexploitation of peat

Line 31: remove: the

Line 32: what process are you referring to? Natural bog drainage? Please specify

Line 46: characteristic

Line 50: The peat layer plays an important role in C sequestration, water storage, and prevention of surface desiccation.

Line 59-63: split into two sentences

Line 68: act as wicks

Line 69-70: significant impact on water

Line 74-76: Species-specific canopy traits could be plastic and change according to the habitat or alternatively, canopy traits could influence the habitats that a given species colonise.

Line 90-93: split into two sentences

Line 94: many imaging techniques require

Line 101: Here, we…

Line 102: shoot density or canopy density? Previously you have only referred to canopy density

Results

Line 134: low amounts… as they might obstruct

Line 138: were collected

Line 147: as a python file.

Line 148: used for image processing

Line 150: define ROI

Line 153: consistently sized

Line 157: define HSV S

Line 158: remove as much non-moss…

Line 159: define: YCrCb Cb

Line 161: Sphagnum

Line 161- 162: I suggest you tell the reader what the function does in more detail, the function name and package are somewhat irrelevant if we don’t understand what is actually being done. What are the local maxima based on (YCrCb Cb values)?

Line 266: scored

Line 285-315: As with my previous review, I still find this section too long. And in agreement with the reviewer 1, this section could be easily simplified and only the most relevant results presented. I really don’t think it is relevant to present all three annotation channels and five threshold channels. I cannot pull out of figure 3 the justification/result presented in Line 311-315. In the end, you don’t use F-measure exclusively to choose the threshold/annotation channels. But I do not understand what else you take into account to come to HSV S and YCrCb Cb (Standard deviation? Range?). Is this something that others will need to decide if they employ the pipeline on their own data? Or are you saying that these two channels should be used exclusively in the pipeline? This does not come across clearly. If this is a step/decision others will need to make, you need to make it much clearer how the decision should be made.

Line 313: annotation channel or threshold channel?

Line 338: use pipeline for consistency rather than software

Figure 5. how many photos (n) does this represent?

Figure 6. same as above

Author Response

We would like to thank the reviewer for their careful consideration of our manuscript. We addressed all their comments and suggestions in this revised version to the best of our abilities.

After consulting with the editor, and as the reviewer had originally suggested, we have reorganised the Material and Methods and Results sections. The Material and Methods section is now placed after the Introduction and before the Results. In the Material and Methods section we described: 2.1. Site selection, 2.2. Field image acquisition, 2.3. Image processing pipeline, 2.4.Thresholding channel selection, 2.5. Pipeline validation and 2.6. Statistical analysis. In the Results section we describe: 3.1. Development of an image processing and analysis pipeline (including 3.1.1. Pre-processing pipeline, 3.1.2. Thresholding optimisation, 3.1.3.Thresholding and annotation channel selection, 3.1.4. Blob detection and capitula counting), 3.2. Estimation of pipeline accuracy and 3.3. Estimation of capitula density in field images. We do believe that this new structure makes the manuscript clearer and easier to follow and we thank the reviewer for their helpful suggestions.

With this new structure, the majority of the thresholding channel descriptions were moved to the Material and Methods section. Figure 3, where we show the F-measures for all the channel combinations was moved to supplemental and section 3.1.3 was simplified (including V2:Line 285-315).  We have also removed the repeated information in section 2.2 in V2, now section 3.1 as requested.

We have also addressed all the other comments as requested.

We provide a revised version with tracked changes but as there was a substantial rearrangement of the sections and the track changes annotation is confusing at times, we also provide a ‘all changes accepted’ version.

We hope that after this extensive and careful revision you are able to accept this manuscript for publication.

Round 3

Reviewer 2 Report

After a second round of reviews and reorganization, I feel that the manuscript reads much better and the message is clear. I congratulate the authors on an interesting and worthy contribution to near-field remote sensing of important peatland ecosystems.